# Distributional Regression Techniques in Socioeconomic Research on the Inequality of Health with an Application on the Relationship between Mental Health and Income

**DOI:** 10.3390/ijerph16204009

**Published:** 2019-10-19

**Authors:** Alexander Silbersdorff, Kai Sebastian Schneider

**Affiliations:** 1Economics Faculty, Georg-August-Universität Göttingen, 37073 Göttingen, Germany; 2Department of Clinical Psychology, PFH Private University of Applied Sciences, 37073 Göttingen, Germany; kschneider@pfh.de

**Keywords:** health inequality, mental health, distributional regression, generalized additive models of location scale and shape, income, inequality measurement, socioeconomic inequality of health, regression, I14, C13, C21

## Abstract

This study addresses the much-discussed issue of the relationship between health and income. In particular, it focuses on the relation between mental health and household income by using generalized additive models of location, scale and shape and thus employing a distributional perspective. Furthermore, this study aims to give guidelines to applied researchers interested in taking a distributional perspective on health inequalities. In our analysis we use cross-sectional data of the German socioeconomic Panel (SOEP). We find that when not only looking at the expected mental health score of an individual but also at other distributional aspects, like the risk of moderate and severe mental illness, that the relationship between income and mental health is much more pronounced. We thus show that taking a distributional perspective, can add to and indeed enrich the mostly mean-based assessment of existent health inequalities.

## 1. Introduction

The relationship between income and health is one of the most widely researched issues in health economics and epidemiology, and has seen countless articles discussing its nature. To this end, various health measures have been employed from objective measures, such as life expectancy [1] and physiological outcomes [2,3] to subjective measures, such as single-item measures [4,5] or composite health scores, like the SF-12 [6] (The SF-12 [7] is a short version of the original SF36 [8]). In terms of the statistical methodology, we can equally observe an array of approaches.

### 1.1. A Review of the Methodological Literature

Most of the heavy empirical ploughing has been carried out by two kinds of workhorse methods. On the one hand, many studies have employed concentration curves and indices [9,10,11,12]. On the other hand, health outcomes are contrasted between different groups with varying incomes (and potentially other covariates) by means of relative frequencies [13], odds ratios [14], or other effect sizes derived from regression estimates [15,16]. This is mostly done by employing conventional regression approaches from the framework of generalised linear models.

Recent years have seen numerous methodological advances, while improved data availability and increased computational capacities have made their application to the analysis of socioeconomic inequalities in health feasible. For concentration curves and indices, the use of level-based concentration curves, instead of rank-dependent ones, have been proposed [17]. In the statistical literature, numerous regression techniques have been developed that go beyond the still-dominant standard generalised linear models in the epidemiological literature, like quantile regression [18], expectile regression [19], conditional transformation models/distribution regression [20,21], recentered influence functions [22], and generalised additive models of location, scale, and shape/structured additive distributional regression models [23,24]. This set of approaches that contemplate not only the conditional expectation but further distributional aspects has already found its ways in the literature on socioeconomic inequalities in health [25,26,27,28], but much of the potential of the application of these methodological advances is still dormant.

### 1.2. A Review of the Literature on Income and Mental Health

A lot of research exists on the relationship between income and mental health, and this section aims to briefly summarize the important findings. One should keep in mind that the research in the field differs in important aspects, such as in the used definitions of mental health (i.e., the presence of mental disorders, continuous scores for constructs like life satisfaction and emotional well-being), the operationalization of income (absolute income, relative income, changes in income, income subsumed in variables like socioeconomic status), or in the subject of interest (individuals or groups).

The prevalence of mental disorders is found to be higher among individuals with lower income [29,30] or among those with lower socioeconomic status [31,32,33]. Equivalently, increased odds for mental disorders are found for individuals with lower levels of income [14] or socioeconomic status [13,32,34,35,36]. Wirtz et al. [37] reported that a higher income is associated with higher MCS scores  (higher MCS scores indicated good mental health, and vice versa), whereby MCS scores are a composite mental health score which we also use in this publication (see Section 2). Additionally, Wood et al. [38] showed that low income predicts mental distress since it functions as an indirect proxy for social rank. McMillan et al. [39] did not find a relation between income and mood or anxiety disorder, but did find one with drug abuse and suicide attempts. The latter was also found by Lee et al. [40]. Sareen et al. [41] revealed that having a low income was a financial barrier for accessing mental health services.

Weich and Lewis [42] found that unemployment and measures of poverty (which were built amongst other variables with the help of household income and savings) had predictive ability for the prevalence of mental disorders, but ongoing subjective financial strain seemed to be an even better predictor for the maintenance and onset of mental illness. Lorant et al. [43] researched the direction of effects, and found that income reductions, as well as increasing financial strain, increased the risk of showing depressive symptoms. On the other hand, increasing income, as well as reduced financial strain seemed not to be related to decreasing depressive symptoms.

Income inequality has also been found to be associated with mental illness. In a systematic review, Pickett and Wilkinson [44] published evidence for a strong association between income inequality and mental disorders, whereas a meta-analysis by Ribeiro et al. [45] revealed small effect sizes. Suicide could be an exception and seems to be more common in more equal societies [46].

Lucas and Schimmack [47] summarized that the correlation between subjective well-being and income is often found to be very small, yet small correlation coefficients can turn into large mean differences when comparing different levels of income. Matz et al. [48] found that the degree of matching between an individual’s spendings and his/her personality had better predictive ability for life satisfaction than income. Even though these effects were found to be statistically significant, the effects may be too small to be relevant [49]. Kahneman and Deaton [15] found that life satisfaction and emotional well-being seemed to rise with increasing income, with emotional well-being satiating at an annual household income of 75,000$ (USA). Additionally, Kushlev et al. [16] reported that a higher income was related with less daily reported feelings of sadness, but not with more daily reported feelings of happiness. Westerhof and Keyes [50] found significant effects when predicting mental illness with the help of income and other socio-demographic variables.

Boyce et al. [51] found the income rank to be more important for the evaluation of life satisfaction compared to absolute income. Yu and Chen [52] found that relative, as well as absolute income was related with negative aspects of emotional well-being, whereas only relative income was related to positive mental well-being. In contradiction to these findings, Sacks et al. [53] provided a review in which they stated that subjective well-being rose with increasing income. According to the authors, this holds for within-country (individual level) and between-country comparisons. The satiation thesis is questioned in this study, and absolute income is identified as more important than relative income.

Overall, we find that there is strong evidence for the assumption that income is related to constructs related to poor mental health, as well as constructs related to good mental health. The current state of research is characterized by disunity about the working mechanisms of the effects of income. On the other hand, there is high methodological cohesion which either sees assessments on the basis of ranks, or is based on arithmetic means.

### 1.3. Aims and Structure of the Paper

This paper aims to add to the young and evolving branch of literature of distributional regression approaches in health economics in two important ways. Firstly, it considers the relationship between income and mental health in a distributional framework. Secondly, it constitutes an exemplary application of the frequentist framework of generalised additive models of location, scale, and shape to the analysis of socioeconomic inequalities in mental health that is designed to guide applied researchers in the use of this powerful, but in some ways, also challenging statistical methodology. In this publication, we focus on the frequentist framework of generalised additive models of location scale and shape. Readers interested in the Bayesian approach of structured additive distributional regression are referred to Klein et al. [24], Silbersdorff [54], and Silbersdorff et al. [27].

The remainder of this publication is structured as follows: Firstly, we explain the mental health concept, variables, and data on which our analysis relies. Then we go on to provide a general discussion on the use of a distributional regression framework and contrast it to the conventional regression approach. In Section 3, we consider the association between mental health and income found in our data. Lastly, we draw conclusions obtained by our analysis. We thereby outline which findings are particularly generated by taking a distributional perspective.

## 2. Materials and Methods

### 2.1. Data

To investigate the relationship between mental health and income, we used data from the German Socio-Econocmic Panel (SOEP) [55]. The SOEP is a “[...] widely used long-running household panel study that seeks to provide a representative view of the entire population [...]” ([55], p. 1). We used only cross-sectional data for the year 2014. The survey for 2014 contained data from 16,037 households [56]. The SOEP data contains a large array of socio-demographic information and various income variables, as well as variables on individual health. Concerning the socio-demographic variables used in our analysis, we followed Silbersdorff et al. [27] and considered the age, marital status, nationality, educational attainment, and place of residence, as well as the household income. Additionally, we incorporated the employment status, as well as the place of living in terms of urbanity in our analysis, since both variables are common in the field of mental health research (i.e., [34,57,58,59,60,61,62]). Further information on these variables, as well as their SOEP identifiers, are provided in Appendix A.1.

In contrast to Silbersdorff et al. [27], we considered a mental health score, rather than a physical health score as the outcome variable. Specifically, we used the Mental Component Scale (MCS) score obtained from the SF-12v2 questionnaire, which also contains the Physical Component Scale (PCS) score used by Silbersdorff et al. [27]. We see this measure as well-suited for our purpose for several reasons. Firstly, MCS scores claim to cover the whole spectrum of self-rated mental health—the MCS scores are designed to have a mean of 50 and a standard deviation of 10 (norm population), where higher scores indicate better mental health conditions compared to lower values [63]. According to Ware et al. [8] (p. 72), very high MCS scores represent an “absence of psychological distress and limitations in usual social/role activities due to emotional problems; (mental) health rated excellent”, whereas very low MCS scores represent “frequent psychological distress; substantial social and role disability due to emotional problems; (mental) health in general rated poor”. Secondly, the MCS scores are a composite score of several items covering the concepts of mental health, social functioning, vitality, and emotional aspects [64], and therefore cover mental health in its multifaceted nature. Thirdly, the MCS scale has good test-theoretical properties. Its internal consistency can be evaluated as acceptable [37], while its convergent and discriminatory validity, as well as its reliability can be evaluated as good [37,65,66]. Lastly, despite the ongoing discussions concerning the definition and measurement of mental health [67,68,69,70,71,72], we see this measure in line with the widely accepted definition of the World Health Organization (WHO) [73]: “mental health is a state of well-being in which every individual realises his or her own potential, can cope with the normal stresses of life, can work productively and fruitfully, and is able to make a contribution to her or his community”. For further information on the MCS scale, also see Appendix A.1.

It should be noted at the outset that differential item functioning by education, age, and sex have been observed for the mental component score compared to the physical component score [74,75]. We condition on these variables to address this drawback.

A drawback of the data gathered by the SOEP is that it fails to include the institutionalized population. Therefore, the results cannot be seen as representative of the whole German population, but only of those outside any institutions—meaning that the results we portray with respect to the risk of very low self-rated mental health are presumably underestimated to some degree.

However, despite this drawback, we deem the analyses on socioeconomic inequalities with respect to mental health to be too important to be left void due to data-related deficiencies, and thus will pursue the following analyses on the basis of this somewhat imperfect data.

### 2.2. Conditional Health Assessment beyond the Mean

In order to facilitate the understanding of the distributional regression approach, which we propose in this publication, we aim to contrast it to the classical mean regression in an intuitive manner.

Let us consider the conditional health distribution for the simplified case, where we only regress MCS on the household income and omit other variables for the sake of simplicity.

In Figure 1, we display the empirically observed conditional mental health distribution for both men and women with a household income of around 15,000 Euro and 30,000 Euro, respectively, with the help of histograms, and contrast it with the obtained estimates from standard mean regression and distributional regression techniques. While the portrayed estimates are estimated for exactly 15,000 Euro and 30,000 Euro, the portrayed empirical distributions are drawn from individuals with incomes in between 12,500 Euro and 17,500 Euro, as well as 27,500 Euro and 32,500 Euro, respectively.

Classical mean regression—including generalised linear models (GLM)—yields information on the conditional arithmetic mean (displayed by the blue dot). This estimate is ideally suited for a comparison of the expected health outcome. In our simplified example, one could thus deduce the expected mental health score of an individual for any income level. From a mean regression-based analysis, we could thus infer that the expected mental health level with a low income is roughly 1.2 units lower than that of an individual with a high income. As is argued by Silbersdorff et al. [27], drawing a conclusion on the basis of this measure is somewhat problematic, as it gives equal emphasis to improving the health of the already healthy, rather than improving the health of the ill. This is arguably problematic in any form of health-centered analysis, and particularly so in an analysis focused on health inequalities.

Distributional regression, on the contrary, does not focus on a particular measure of the distribution, but directly aims to estimate the whole conditional response distribution for each group (displayed by the red function). From this conditional distribution, any desirable distribution measure can be calculated in principle. One can therefore explore the influence of an explanatory variable on any desired statistical distributional measures like the mean, variance, or skewness, as done by Silbersdorff et al. [27]. Moreover, we can consider the inequality and risk measures associated with that distribution. In our particular case of analysing health inequalities, we can deduce risk measures defined as the share of individuals falling below a certain threshold. For example, we could consider the risk of belonging to the lowest 5% of all individuals in the sample, which is displayed by the shaded red area underneath the distribution to the left of the point *T* (for threshold). Figure 1 shows that the shift in probability mass translates to a smaller probability for falling below the defined threshold for a person with high income. Furthermore, we can deduce the risk for any given income of falling below a globally defined threshold, which can be seen as the threshold to suffering from poor mental health.

While we consider only two risk measures in the following, it should be stressed that, in general, it is straightforward to use the estimated conditional mental health distributions to explore any number and kind of further risk measures—be they further thresholds or other distributional measures, like inequality indices. One major advantage of the distributional approach is thus the need for only one estimation process to estimate a host of distributional measures. This stands in contrast to equally feasible approaches to estimate each distributional measure individually that render an array of problems associated with simultaneous inference on several models [76].

The more comprehensive distributional perspective thus allows for a nuanced assessment of the income–health relationship which not only contemplates expected health but other distributional aspects as well, like risk measures. However, this naturally also implies a much greater model space that is accompanied by some pitfalls, just as the assessment on an array of separately estimated models would be. In the following, we thus consider some vital components of the approach from a user’s perspective, and give some guidelines for safe and sound usage. The underlying baseline recommendation for distributional regression users with limited or no experience is to commence your applied analysis by using simple models, rather than exploiting the extensive statistical artillery that has been made available over the past few years. The guidelines we give here are thus intended to provide a starting point for an applied researcher to use distributional regression in the context of health inequalities and who may want to get a first sense of the existence and magnitude of potential effects beyond the mean.) The estimation is done with the help of the GAMLSS-package [23] for the statistical software R [77].

### 2.3. Considering the Conditional Distribution Set-Up

In the distributional regression framework, the distribution of the response variable—here, the mental health distribution—is described by a distribution that is conditional on a set of explanatory socioeconomic and potentially further variables, i.e., $D(Y|x_{1},\cdots ,x_{K})$.

One assumes a parametric distribution, which parameter values depend on the explanatory variables, that is, $D(\theta_{1}\left(x_{1},\cdots ,x_{K}\right),\theta_{2}\left(x_{1},\cdots ,x_{K}\right),\cdots ,\theta_{L}\left(x_{1},\cdots ,x_{K}\right))$. Therefore, one relates its parameters ($\theta_{1},\cdots ,\theta_{L}$) to the variables ($x_{1},\cdots ,x_{K})$ via a regression predictor, which contains regression coefficients, in the form:(1)$$\begin{array}{cc} g_{l}\left(\theta_{l}\right)= & \;\eta^{\theta_{l}}, \end{array}$$
where $\theta_{l}$ denotes the *l*-th parameter of the distribution, $g_{l}$ denotes the corresponding link function, and $\eta^{\theta_{l}}$ the predictor.

By relating all the parameters of the response distribution to a regression predictor, one models the response distribution conditional on the variables. After the estimation, the variables can be fixed at specific values to define groups that are of special interest. The resulting conditional response distributions can then be compared between the groups using different measures (i.e., risk measures, see Section 3.3.1) describing certain aspects of the conditional response distribution.

The analysis of socioeconomic inequalities using distributional regression requires the consideration of various modelling components, which we will discuss in the following.

#### 2.3.1. The Conditional Health Distribution

The greatest strength, and potentially the greatest weakness of the distributional regression lies in the use of a parametric distribution for modelling the health distribution. If the selected parametric distribution provides a sufficiently good approximation to the response under consideration, the assumption will facilitate the estimation to a substantial degree. Choosing a distribution that provides a sufficiently close approximation to the response distribution has thus been found to provide more stability in the estimation process, meaning, among other things, higher estimation precision and smaller standard errors [78]. For the usually rather limited sample sizes available in health inequality research, this is of particular importance for the assessment of inequality in the tails, as data there is naturally very scarce. If, however, the selected distribution does not sufficiently approximate to the response distribution, the outcome will be very detrimental to the model fit, and may potentially yield misleading results. The choice of a good parametric distribution is thus key to any analysis employing a distributional regression framework.

One major problem with health scores like the MCS is that they follow a shape (negative skewness) that runs counter to most usual parametric distributions. Thus, Silbersdorff et al. [27] suggests the following linear transformation:(2)$$g_{MCS}\left(S\right)=S^{*}=\frac{S_{0}-S}{S_{scale}},$$
where *S* is short for score and denotes the untransformed MCS scores, and $S^{*}$ denotes the transformed MCS scores. $S_{0}$ is a constant ensuring that $S^{*}$ has positive support, and $S_{scale}$ is a rescaling factor. Within this publication, $S_{0}=100$ and $S_{scale}=10$ are used such that the distribution of $S^{*}$ is positively skewed and the values are restricted to the interval [0,10]. A contrast between the original and the transformed marginal distribution of the MCS score is shown in Figure 2.

Using this transformation, we have done extensive comparisons based on information criteria and residual diagnostics on a host of potential distributions. A distribution is an option if it supports the permissible range of mental health values, here being [0,10]. For an extensive account, see Appendix A.2. Here, we concentrate on three distributions—the gamma distribution (Ga) with two parameters linked to a regression predictor, the Box–Cox power exponential (BCPE) with three out of four parameters linked to a regression predictor, and the generalised beta distribution of the second kind (GB2) with all four parameters linked.

The comparison on the basis of various information criteria shows that the more complex three- and four-parameter distributions generally outperform the more simplistic gamma distribution with only two parameters.

Residual diagnostics reveal that this difference is due to the rigidity of the gamma distribution in the tails, which leads to inferior fits. Residual diagnostics plots are provided in Appendix A.3. However, the observed differences are minor, and all three distributions provide adequate fits in the sense that from the most flexible four-parameter distribution to the more rigid two-parameter distribution, all distributions yield similar results concerning the risk measures used for the assessment of socioeconomic inequalities in health.

Moreover, the more complex characteristics also feature various problems—first and foremost, there is decreased estimation stability due to the much increased model complexity. This not only leads to much wider confidence intervals, but also to some undesired statistical artifacts, like very high expected mental health outcomes for very low incomes.

With regard to the choice of the response distribution, we thus conclude that any of the three distributions provide a sufficiently good fit to model health outcomes in the distributional framework we employ here. While all three distributions are thus viable candidates to be used in the analysis of socioeconomic inequalities in health, we advise inexperienced users to stick to the simplest of the three distributions, namely the gamma distribution.

#### 2.3.2. The Predictor Specification

The predictor specification defines the nature of the dissection of the population into the groups used for the analysis. In principle, all parameters are allowed to have separate predictors containing potentially completely different variables and structures (e.g., linear interaction terms, splines, local regression smoothers, ridge and lasso regression terms, neural networks, ...) are implemented in the GAMLSS software [23]. However, by using simple linear predictors, the groups’ differences are coerced to follow certain patterns that facilitate the estimation procedure, stabilize the estimation results, and potentially allow for a straightforward interpretation of the results. For the sake of simplicity, model comparability, and technical feasibility, we thus recommend using one generic straightforward linear predictor for all the distributions’ parameters that takes the form:(3)$$\begin{array}{cc} \eta^{\theta_{l}}= & \;\beta_{0}^{\theta_{l}}+\sum_{k=1}^{K}\beta_{k}^{\theta_{l}}x_{k}. \end{array}$$

Even in this simplified predictor framework, it must be noted that the vector of all regression coefficients β entails parameters not only for one predictor, but for all *L* predictors required to specify the response distribution yielding L×(K+1) parameters, which can quickly yield a daunting model complexity for the kind of finite datasets and computational capacity that are available for research in health inequalities.

Thus, the use of distributional regression requires researchers to heed the advice of Box [79] to select variables economically and refrain from any form of “kitchen sink” regression, with the obvious difficulties for causal inference.

#### 2.3.3. The Link Function and Other Technical Aspects

The link function $g_{l}$ links the predictor to the parameters of the response distribution. It is first and foremost designed to ensure that the parameters are constrained to valid values. Moreover, they can transform the impact of the covariates on the parameter. While the effect of applying different link functions is far from negligible, we found that the default choices (see Table A1 of Appendix A.2) generally yielded reliable results (pending the quality of the distribution).

Equally, the additional inputs that are usually required (e.g., optimization options) are generally not critical, as long as one does not veer too far from the sensible options usually put down as a default, and as long as one sticks to simple two-parameter distributions and linear predictors. However, once the models get more intricate as many-parameter distributions or complex predictors are selected, the optimization routines will quickly run into the curse of dimensionality and require thoughtful usage.

Our bottom-line suggestion for the applied research on socioeconomic inequalities in health using distributional regression techniques is thus to keep things simple, in the sense that simple distributions (like the two-parameter gamma distribution) and simple predictors (like the linear predictor) should be employed. While more complex distributions and predictors may often yield models with better fit, the risk of technical or even inferential problems (if the models are not applied with the necessary care and background knowledge) does, in practice, outweigh the theoretical benefits.

The estimation of GAMLSS, as provided by the GAMLSS software [23] is based on the maximum likelihood principle [80]. For the sake of simplicity, assuming that no smoothing functions, random effects, or modeling techniques are present in the model, estimates are obtained by maximizing the log-likelihood defined by $\ell =\sum_{i=1}^{N}ln\left(f\left(y_{i}|\mu_{i},\sigma_{i},\nu_{i},\tau_{i}\right)\right)$, where each parameter is related to a regression predictor containing the parameters to be estimated [81]. For more complex modelling structures, estimates need to be obtained by maximizing a penalized log-likelihood given by $\ell_{p}=\ell -\frac{1}{2}\sum_{k=1}^{4}\sum_{j=1}^{J_{k}}\gamma_{kj}^{T}G_{kj}\left(\lambda_{kj}\right)\gamma_{kj}$ [81]. The penalized log-likelihood is defined under the assumption that the model can be written as a random effect model with smoother s(x)=Zγ, where Z is a design matrix constructed with the data (*x*) and γ is a parameter vector to be estimated under the restriction of a quadratic penalty $\gamma_{kj}^{T}G_{kj}\left(\lambda_{kj}\right)\gamma_{kj}$, with penalty matrix G and hyperparameters λ that regulate the amount of smoothing [80,81]. The optimisation is carried out using a two-cycle backfitting algorithm. The user of the GAMLSS software can choose between two specific algorithms: The RS and CG algorithm [23]. While the RS algorithm is more stable and faster in general, the CG algorithm may outperform the RS algorithm in situations where the distribution parameters are highly correlated [81]. Using only linear predictors and the gamma distribution, the computational stability of the RS algorithm tends to outweigh the advantages of the CG algorithm, so we recommend using the former as a default option to inexperienced users. Detailed information on the estimation routines are provided by Stasinopoulos and Rigby [23], Rigby and Stasinopoulos [80], and Stasinopoulos et al. [81].

## 3. Results

Following the notions from the previous section, we will use the two-parameter gamma distribution to model the transformed MCS in a distributional regression framework. In this section, we solely display and discuss extended results (regression estimates, distributional measures, exemplary conditional densities) for this distribution, but also provide other distributions (i.e., the BCPE and GB2 distribution) in the Appendix A.4.

### 3.1. The Predictor

The following regression predictor set-up is applied to all parameters of the distribution, with θ denoting a generic parameter representing either of the gamma distribution’s parameters, μ or σ:(4)$$\begin{array}{cc} \eta_{\theta }=\beta_{0}^{\theta } & +\beta_{1}^{\theta }AGE+\beta_{2}^{\theta }AGESQ+\beta_{3}^{\theta }LOGINC+\beta_{4}^{\theta }GER\\ & +\beta_{5}^{\theta }EDU_{2}+\beta_{6}^{\theta }EDU_{3}+\beta_{7}^{\theta }EDU_{4}+\beta_{8}^{\theta }MAR_{2}\\ & +\beta_{9}^{\theta }MAR_{3}+\beta_{10}^{\theta }MAR_{4}+\beta_{11}^{\theta }EAST+\beta_{12}^{\theta }CITY\\ & +\beta_{13}^{\theta }UNEMPLOYED, \end{array}$$
where AGE denotes the age of the individual in years, and AGESQ denotes the squared age of the individual. LOGINC refers to the logarithm of the annual net equivalised household income. GER refers to a dummy variable indicating whether an individual is a German national or not. The variables $EDU_{1}$–$EDU_{4}$ represent the respondents’ educational attainment measured with the International Standard Classification of Education (ISCED97) [82]. $EDU_{1}$ includes all individuals with ISCED levels 0, 1 and 2, representing pre-primary, primary, and lower secondary education; $EDU_{2}$ includes all observations with ISCED level 3 representing upper secondary education; $EDU_{3}$ includes all observations with ISCED levels 4 and 5; $EDU_{4}$ includes all observations with ISCED level 6. In Germany, this means that the first education level ($EDU_{1}$) contains the educational attainment of finishing kindergarten and/or Haupt-, Realschule, or the Gymnasium (ohne Oberstufe). The second education level ($EDU_{2}$) contains the educational attainment of finishing Berufsschule, Gymnasium (Oberstufe), or equivalent schooling levels. The third education level ($EDU_{3}$) contains the educational attainment of being awarded a Bachelor’s or Master’s degree. The fourth education level ($EDU_{4}$) contains the educational attainment of being awarded a doctorate or a habilitation. The variables $MAR_{1}$–$MAR_{4}$ represent the respondents’ marital status, with $MAR_{1}$ containing individuals who are married and living together or who are in a same-sex partnership and living with their partner; $MAR_{2}$ contains all individuals who are married but living separately, divorced, having a dissolved registered partnership, or those who have a registered same-sex partnership but are living separately; $MAR_{3}$ contains all individuals who are single; and $MAR_{4}$ contains all individuals who are widowed. EAST denotes a dummy variable indicating whether an individual lives in former West or East Germany. CITY denotes a dummy variable indicating whether an individual lives in an urban area or not. UNEMPLOYED denotes a dummy variable indicating whether the respondent is unemployed or not.

Note that the predictor does not contain coefficients for the variables $EDU_{1}$ and $MAR_{1}$ since they function as the reference group, meaning their effect is subsumed in the intercept $\beta_{0}^{\theta }$. Further information on how the above mentioned variables were built, as well as their theoretical background and SOEP references, are provided in Appendix A.1.

### 3.2. The Regression Coefficients

The results of the regression are displayed in Table 1, with the standard errors denoted in parentheses following the coefficients’ estimate. The standard errors were computed on the basis of the variance–covariance matrix provided by the GAMLSS software [81]. It should be noted that these values are to be interpreted with caution. For this reason, we provide the more reliable bootstrap-based intervals for the analysis of the mental health and income association.

Due to the intricate nature of the parameter interpretation, we have refrained from discussing the results at length. The interpretation is intricate, since the parameters of the response distribution cannot necessarily be equated with a distributional measure, like the expectation, variance, or skewness (which would be more or less straightforward to interpret). Distributional measures are often functions of several parameters of the response distribution. We used the gamma distribution in mean parameterisation with density:(5)fY(y|μ,σ)=y1σ2−1e−y(σ2μ)(σ2μ)1σ2Γ(1σ2),
for y>0, μ>0, σ>0, with E(Y)=μ and $Var\left(Y\right)=\mu^{2}\sigma^{2}$.

It should, however, be pointed out that living in separation of a partner ($MAR_{2}$) has a highly significant coefficient for σ, indicating variations beyond the expected value of the distribution. Similar observations—to various degrees of significance—can be made for being unemployed, as well as for having received a higher education ($EDUC_{4}$). Last but not least, the household income (LOGINC) features highly significant effects on parameters for both men and women, and will be considered in more detail in the following.

### 3.3. A Distributional Perspective on the Association between Mental Health and Income

In order to assess the relationship between mental health and income, while accounting for the other variables, we followed Silbersdorff et al. [27] and employed the concept of “average Joe” and “average Jane” representing an ideal-typical man and woman, respectively, with average characteristics. In our case, the following variable combination was assumed: 52 years old, married, living in West Germany, having standard secondary education, having German nationality, being employed, and not living close to a city centre. In the first step, we consider the relationship between three distribution measures and income at large, and subsequently consider the resultant differences between 15,000€ and 30,000€ in more detail.

#### 3.3.1. Visualizing the Mental Health and Income Relationship

Figure 3 shows the relation between income and the different distribution measures for “average Joe” and “average Jane”.

The x-axis begins at 4700€, which roughly matches the income received on the basis of the German Social Security. The regular level of social security for a single person was set at 391€ per month in 2014. Approximately one percent of the survey participants had an income below that threshold, and reaches up to 100,000€ to exclude the economic elite, which is not adequately represented in the SOEP [83]. Furthermore, the x-axis is plotted on the log scale. The dashed lines show 0.95%-pointwise confidence intervals of the portrayed estimates. They were obtained by bootstrap sampling with 2500 bootstrap samples. Details on the bootstrap procedure and recommendations are provided in Appendix A.5.

The graph at the left portrays the expected outcome with varied incomes. The y-axis displays back-transformed MCS scores. It can be observed that the MCS scores increase with increasing income for males and females, as one would expect.

The right column of Figure 3 shows the effect of income on the conditional probability that a person will fall below specific threshold values *T* of the MCS scale. Specifically, we used the threshold of having a MCS score below the lowest quintile (denoted by $R_{0.2}$) and the lowest vingtile (denoted by $R_{0.05}$). Note that we used separate thresholds for men and women, with $T_{0.2}^{♂}\approx 45$, $T_{0.2}^{♀}\approx 42$, $T_{0.05}^{♂}\approx 34$ and $T_{0.05}^{♀}\approx 30$. Ware et al. [8] found that among individuals with MCS scores between 30–34, 89% reported symptoms related to depression, while this was 59% for individuals with MCS scores between 40–44. These risk measures are essential in our analysis that should take a perspective that goes beyond the mean. Modeling the whole conditional response distribution gives the possibility of deriving findings extracted from the resulting probability density functions. The idea of the chosen risk measures is to portray the risk of minor or major mental health issues (with $R_{0.2}$ entailing both, and $R_{0.05}$ only the latter).

Both risk measures show that the risk of falling below the respective thresholds decreases with increasing income. In particular, one can observe that for $R_{0.05}$, the individuals with very high incomes can practically eliminate their risk of being in a very bad mental health condition, which mirrors the finding from Silbersdorff et al. [27] on the PCS score. The association regarding the distributional measures for the BCPE and GB2 model is very similar, and has been displayed in Appendix A.4.

#### 3.3.2. Contrasting Mental Health for Two Income Levels

Let us now consider the difference between two income levels—15,000€, representing the 25th percentile or the median of the poorer half of the population, and 30,000€, representing the 75th percentile or the median of the better-off half of the population.

Figure 4 displays the estimated conditional probability density functions for average Joe (blue) and Jane (red) at an income of 15,000€ (solid) and 30,000€ (dashed). Additionally, Table 2 shows the corresponding distributional measures.

The portrayed densities reveal that differences are not equal across the distribution, but feature a relocation of probability mass from the lower end of the distribution to the centre of the distribution (when changing from 15,000€ to 30,000€). In other words, the change in income mainly incurs a depletion of the risk of having low or very low mental health scores, and changes only very little for the upper spectrum of the distribution.

Depending on the distribution measure, we thus found relative differences of varying magnitudes. The relative difference is the absolute difference of the measures for 15,000€ and 30,000€,  divided by the measure for 15,000€. For males, the expected mental health outcome increased from 51.86 to 52.56, a relative difference of only 1.3%. For females, the corresponding change was 50.15 to 51.16, yielding a relative difference of 2.0%.

By contrast, considering the risk measure $R_{0.2}$ which focuses precisely on the lower end of the distribution, we saw a change from 0.222 to 0.187 for males, meaning that the average Joe with an income of 15,000€  with a risk of suffering from a minor or major mental health issue was expected to be 15.8% higher than that of a comparable man with an income of 30,000€.

This difference becomes even more pronounced when focusing on the more extreme $R_{0.05}$ measure that is thought to focus on the major mental health issues alone, and features a 35.8% and 40.1% change for men and women, respectively.

These results show that, like for PCS, the association between income and health is much more pronounced at the lower end of the health spectrum than it is for expected health.

## 4. Conclusions

In this publication we have provided some general insights into the use of distributional regression approaches on the issue of socioeconomic health inequality in general, and provided an application to the relationship between mental health and income in detail.

This publication complements the findings related to physical health provided by Silbersdorff et al. [27]. Taken together, both publications show that taking a distributional perspective can reveal insights that otherwise would not have been revealed. While Silbersdorff et al. [27] utilised a distributional regression approach within the bayesian framework, in this publication the frequentistic framework was followed. Independent of the framework, the aim was to estimate complete conditional (mental) health distributions rather than obtaining single estimates for the expected outcome. These provide more detailed information on the relationship between income and (mental) health, which we made explicit by defining risk measures. Regarding this relationship between mental health and income, the risk measures showed that income is more strongly related to the risk of being in a poor mental health condition than to the expected mental health. Thus, this publication contributes to the literature stated in Section 1.2, since it disentangles the effect of income on mental health and gives a first indication that the expectation-based perspective may underestimate the importance of income, not only for physical health, but also in mental health.

Regarding methodological guidelines, we proposed to use simple distributional regression models with linear predictors and distributions with few parameters, like the gamma distribution with two parameters. Additionally, we proposed to use simple and intuitive measures to exploit the information generated by estimating full conditional response distributions. Subsequently, we focused on risk measures on the basis of this distribution, but other measures could also be considered. Looking also at more complex models, we find that while these often provide better model fits, they also feature problematic fits in the sparsely populated areas of the covariate space, as well as much wider confidence intervals due to the higher model instability. Following Box [84] it should be noted that frequentist models, as well as bayesian models suffer from under-accounting uncertainty due to the unconditional assumption of the model specification. The wider confidence intervals of the more complex models thus go some direction in correcting for this under-accounting, but do so only implicitly.).

Naturally, the proposed methods can be extended even further. For example, one could make use of the longitudinal nature of the data, yielding a deeper data foundation to the analysis [14]. The GAMLSS framework theoretically provides the required statistical repertoire, as random effects can be incorporated [81] and longitudinal approaches have been applied in other fields [85,86]. One could overcome the rather questionable separation between physical and mental health [87] and model them jointly by bivariate distributional regression, which is currently being developed [88]. While these extensions may provide interesting research avenues for the methodologically minded health inequalities researcher, we believe that for many applied researchers, a simple distributional regression approach will suffice to gain sound and interesting insights into the matter.

Even if using a relatively simple predictor and distribution with few parameters, it must be noted that the modelling of distributional regression is still a complex undertaking, yielding mostly indicative and approximate, rather than robust and precise results. Further evidence from theory and other modelling approaches is thus usually needed for any causal conclusions to be made.

Nonetheless, following Chalmers [89], new methods generating new empirical evidence on persistent questions are sometimes needed to generate progress in science. The distributional regression approach allows for looking at the full health distribution while conditioning on a set of variables, and therefore provides additional perspectives that should be considered for any comprehensive assessment on the much-discussed relationship between income and health.

## Figures and Tables

**Figure 1 ijerph-16-04009-f001:**
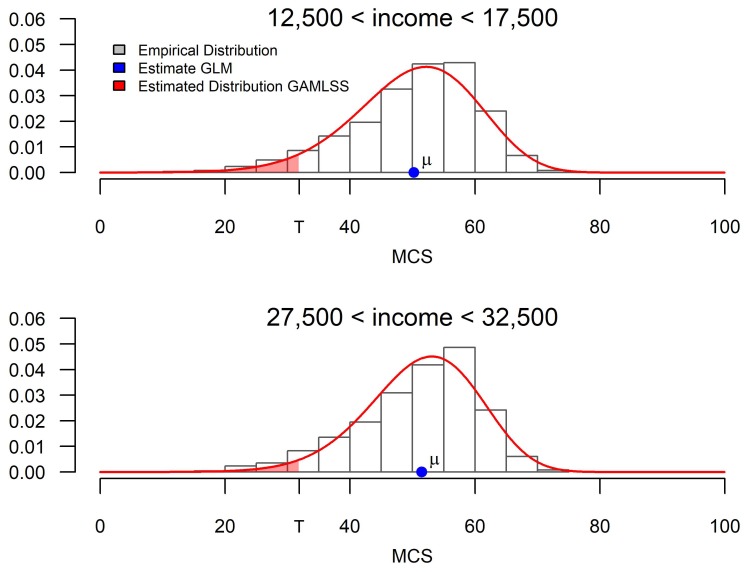
Comparison of estimates between GLM and GAMLSS.

**Figure 2 ijerph-16-04009-f002:**
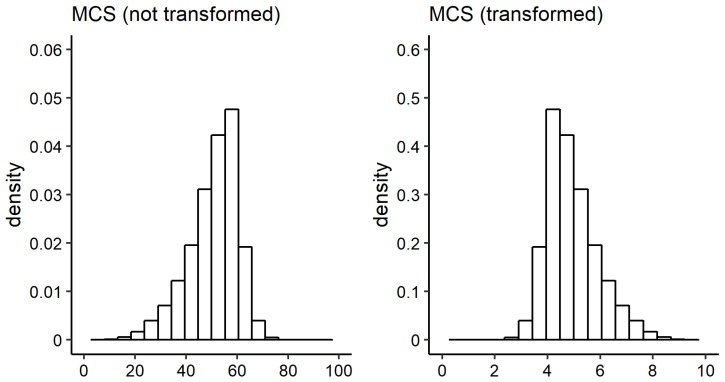
Histogram of MCS scores. Left: not transformed, right: transformed.

**Figure 3 ijerph-16-04009-f003:**
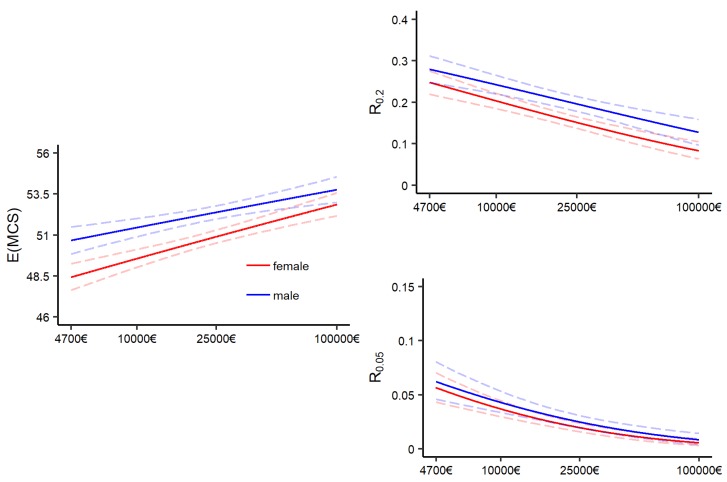
Effect of income on distribution measures for the GA model: Ga (μ = 🗸, σ = 🗸). **Left**: Effect of income on the expectation. **Right**: Effect of income on risk of falling below the lowest quintile and lowest vingtile.

**Figure 4 ijerph-16-04009-f004:**
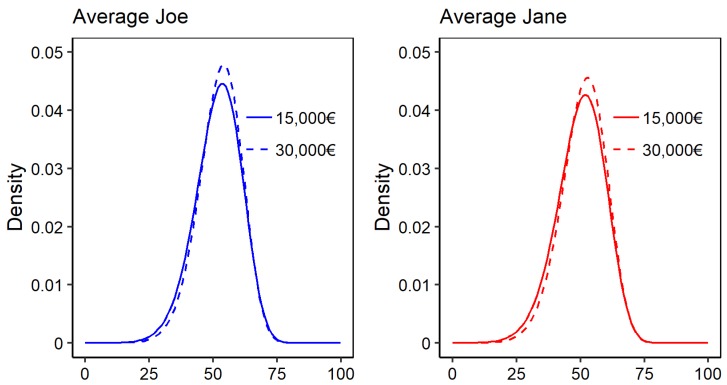
Estimated conditional density of average Joe (blue) and Jane (red) at income levels of 15,000€ and 30,000€.

**Table 1 ijerph-16-04009-t001:** Linear effects of $\eta^{\mu }$ and $\eta^{\sigma }$ for MCS in the Gamma model, with standard errors in parentheses.

	Male				Female			
	$\eta^{\mu }$		$\eta^{\sigma }$		$\eta^{\mu }$		$\eta^{\sigma }$		
const.	1.771 ***	(0.045)	−1.003 ***	(0.172)	1.944 ***	(0.043)	−0.994 ***	(0.158)
MAR_2_	0.018 *	(0.008)	0.115 ***	(0.028)	0.031 ***	(0.006)	0.075 ***	(0.022)
MAR_3_	0.007	(0.006)	0.052 *	(0.025)	0.021 ***	(0.006)	0.042 .	(0.022)
MAR_4_	0.043 **	(0.014)	0.036	(0.048)	0.025 **	(0.009)	0.028	(0.03)
GER	−0.019 **	(0.007)	−0.028	(0.027)	−0.032 ***	(0.006)	−0.035	(0.025)
UNEMPLOYED	−0.026 **	(0.010)	−0.048	(0.034)	−0.027 **	(0.009)	−0.094 **	(0.029)
EDUC_2_	−0.007	(0.007)	−0.046 .	(0.027)	−0.020 ***	(0.006)	−0.038 .	(0.021)
EDUC_3_	−0.004	(0.009)	−0.052	(0.032)	−0.002	(0.008)	−0.006	(0.028)
EDUC_4_	−0.008	(0.008)	−0.101 ***	(0.031)	−0.016 *	(0.007)	−0.056 *	(0.026)
CITY	0.014 ***	(0.004)	0.012	(0.016)	0.005	(0.004)	−0.013	(0.015)
EAST	0.008 .	(0.005)	−0.036 .	(0.019)	0.003	(0.005)	0.001	(0.017)
AGE	0.003 ***	(0.001)	0.005	(0.003)	0.001	(0.001)	0.001	(0.003)
AGESQ	0.000 ***	(0.000)	0.000	(0.000)	0.000 **	(0.000)	0.000	(0.000)
LOGINC	−0.021 ***	(0.004)	−0.083 ***	(0.017)	−0.029 ***	(0.004)	−0.070 ***	(0.015)

*Notes.* ***, **, * and . refer to significance levels with p≤0.001, p≤0.01, p≤0.05 and p≤0.1 obtained from *t*-tests with $H_{0}:\beta_{j}^{\theta }=0$ and $H_{1}:\beta_{j}^{\theta }\neq 0$.

**Table 2 ijerph-16-04009-t002:** Expectation and risk measures for average Joe and Jane at income levels of 15,000€ and 30,000€.

		15,000€		30,000€		Relative Difference	
**male**	μ	51.86	[51.4 ; 52.31]	52.56	[52.14; 52.98]	0.013	[0.008 ; 0.02]
	$R_{0.2}$	0.222	[0.203 ; 0.241]	0.187	[0.168 ; 0.205]	0.158	[0.105 ; 0.216]
	$R_{0.05}$	0.034	[0.027 ; 0.042]	0.022	[0.017 ; 0.028]	0.358	[0.256 ; 0.456]
**female**	μ	50.15	[49.72 ; 50.6]	51.16	[50.74 ; 51.57]	0.020	[0.014 ; 0.026]
	$R_{0.2}$	0.180	[0.165 ; 0.195]	0.141	[0.127 ; 0.156]	0.214	[0.161 ; 0.265]
	$R_{0.05}$	0.028	[0.023 ; 0.034]	0.017	[0.013 ; 0.021]	0.401	[0.311 ; 0.483]

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
