# Peer review of "Distributional Regression Techniques in Socioeconomic Research on the Inequality of Health with an Application on the Relationship between Mental Health and Income"

_ijerph, 2019, doi:10.3390/ijerph16204009_

Round 1

Reviewer 1 Report

See added file.

Author Response

Dear Reviewer,

We would like to thank you for the careful review of the paper, your feedback and in particular for your criticism, which we have to admit is spot on regarding some of the weak aspects of the paper that we acknowledge exist.

Please find our point-to-point reply regarding your remarks in the following.

It is regularly emphasized that “modelling of distributional regression is still a complex undertaking, yielding mostly indicative and approximative rather than robust and precise results” (p. 14). In fact, many of the methodological suggestions are not based on a theoretical reasoning but on considerations of practical convenience. Models with better fits are said to create “the risk of technical or even inferential problems” (p. 9), “the use of distributional regression requires researchers to heed the advice to select variables economically” (p. 8), or “for the sake of simplicity and model comparability, we recommend to use one generic straightforward linear predictor” (p. 8). The same somewhat impressionistic arguments are given for the (crucial) choice of the parametric distribution underlying the analysis. 

With our manuscript we aim to open the door for applied researches for taking a distributional perspective on health inequalities. Thus, our methodological suggestions are to a large extent based on practical considerations regarding the technical application of GAMLSS to the field of health inequalities at large and not the specific inquiry at hand.

To this end, we argue that “simple” distributional models (using a two-parameter distribution as a response distribution with its parameters linked to the same generic linear predictor) are sufficiently well suited for exploratory analyses to get a sense of the magnitude of potential effects beyond the mean. We have tried to show with the help Appendix of A.2 – A.3 (p. 17 – 21) that the application of the approach with the rather simple (two-parameter) Gamma-distribution is sufficiently well-suited to explore effects beyond the mean for the MCS outcome and that it is thus unnecessary to force inexperienced distributional regression users to dive into the often murky and troublesome waters of frequently unstable and sometimes technically problematic more complex distributions with three or even four parameters.

Equally the variables chosen for the predictor are largely based on the based on the variable selection in Silbersdorff et al. (2018), albeit being modified to some extent on the basis of theoretical considerations regarding mental health. However, it should be noted that the GAMLSS approach also allows to incorporate the explanatory variables in many complex different functional forms, i.e. as interaction terms, (fractional, piecewise) polynomials, smooth functions, local regression smoother, ridge and lasso regression terms or even techniques related to neural networks and decision trees. But naturally these more complex structures come at the cost of increased technical complexity and potential pitfalls for the applied researcher – so that we recommend applied researchers to stick to linear predictors (at least at the outset). We have clarified this in the manuscript.

Generally speaking, we believe that the insights gained by using a potentially oversimplified distributional approach with somewhat “impressionistic” predictor and distribution choice can already shed important insights onto the aspects of health inequality under investigation but acknowledge that our reasoning along those lines should be more clear cut in the paper.

Accordingly, we revised our manuscript by …

Giving our analyses and recommendations a more explanatory character (p. 6) Emphasizing that our recommendations should be seen as starting point for modelling aspects of health inequality beyond the mean (p. 6)

The authors do not give sufficient information about the details of the estimation procedure –in my view, references to a computer program (GAMLSS) are not sufficient to understand well how the method works.

Information on the estimation procedure were originally left out intentionally and we restrained ourselves to referencing the existent statistical literature to provide as simple a text as possible. But we recognise that a short overview on the underlying algorithms would probably advance rather than hinder the understanding of most interested readers. We therefore added an overview with the basic principles of the estimation procedure and provided references for further details. The information is given at p. 9 of the revised manuscript.

The practical illustration does not really show that all the work (and the acceptance of convenient assumptions) pays off. What do we gain by distributional regression compared to a standard approach in which we would just estimate a binary (e.g. logit or probit) model for the probability to be “below the threshold of MCS”?

The fundamental advantage of estimating the whole conditional distribution is that we have access to all potentially interesting distributional measures that can be derived from a probability density function. The portrayed risk measures are thus only two possible measure to look at and many more (including Variance, Skewness, etc.) are feasible. We emphasised this gain of using a GAMLSS based on your comment on page 6 of the revised manuscript.

Naturally for the portrayed risk measures the alternative to estimate several logit/probit models for falling below different thresholds does come to mind and is a common procedure in the applied literature. However, it must be noted that a drawback of this approach would be the alpha-error accumulation when it comes to joint inferences for several risk measures. Given the strong relation between estimates for two (or more) risk measures, the correct assessment of the alpha-error estimation is very try. By estimating a distributional model, we only have one estimation procedure and thus do not need the adjustment for the alpha-error. We emphasised this advantage at page 6 of the revised manuscript.

A third and potentially most important advantage is that, if (and only if) the chosen parametric distribution is a sufficiently good approximation to the response distribution, the estimation procedure will be much more stable yielding more precise estimates and smaller standard errors) [2]. Given that the information in the tails of the distributions is usually scarce, such assumptions are often necessary to obtain any form of informative insight from the finite sample sizes we are usually confronted with [2]. We added this argument on page 7 of the revised manuscript.

The restriction to work with a cross-section only (also taken for reasons of feasibility) removes the most obvious possibilities to make causal inferences. Existing techniques (such as logit or probit) can easily be accommodated for panel estimation with individual fixed effects.

This is again a valid comment and is a matter we are currently concerned with. Given that the SOEP data longitudinal data is available a longitudinal GAMLSS is naturally desirable. However, there are a couple of drawbacks, which one must grapple with (and some of which we currently are working on):

The distributional perspective takes the conditional perspective on those aspects that are analytically tangible and takes the marginal perspective on those aspects which are not analytically tangible, explicitly leaving them to the stochastic domain. As elaborated in more detail in Silbersdorff (2017), conditioning on individual specific effects implies that the inequality in health analysed is constrained to the intra-individual inequality of health. [2] Conditioning on the individual, the only variation left to analyse is the variation of health for a specific individual. However, this would put a very different and in our case undesired interpretation on the inequalities in health – since we are analysing inequalities in health at the societal level in the sense that we are considering randomly selected individuals with a given set of covariates and not pre-specified individuals with a given track record in health (and the considered covariates). In order to allow for the contemplation of the latter, we are currently working on a reliable implementation for the incorporation of random effects into the gamlss framework to account for repeated measurements. However, the increased model complexity frequently leads to failures in reaching convergence criteria in the estimation routines. In particular, figuring out how each parameter of the response distribution is affected by individual specific effects turns out to be technical challenge.

Thus we opt for confining ourselves, to state on page 14 of the revised manuscript that it is theoretically possible to estimate longitudinal GAMLSS.

A further question: the authors apply a linear transformation to the raw data (p. 7), changing both the scale and the location of the outcome variable. Inequality (and poverty) measures are sensitive to such transformations. How should this be taken into account in the further analysis? (The problem does not pop up in the analysis in the paper, because the authors do not calculate an overall inequality measure.)

This is a question that we have also discussed internally at length in various contexts. In this case, the transformation is unproblematic though, since the transformation is only “internatally” during the estimation process – essentially the data is flipped and scaled to fit the distribution and then flipped and scaled back to assess the outcome. Any potential inferences (on inequality or the like) are based on the backtransformed data and thus not affected by potential transformation effects.

details

We have fixed all spelling mistakes.

References

[1] Silbersdorff, A.; Lynch, J.; Klasen, S.; Kneib, T. (2018): Reconsidering the Income-Health Relationship using Distributional Regression. Health Economics, 27, 1074–1088.

[2] Silbersdorff, A. (2017). Analysing Inequalities in Germany: A Structured Additive Distributional Regression Approach. Springer.

Reviewer 2 Report

The paper investigated the relationship between mental health and household income by using generalized additive models of location, scale and shape and thus employing a distributional perspective. The topic is interesting and the results are clearly presented. To be published, I have some minor concerns.

The abstract needs to be improved, i.e., providing more insights of this paper and more details of the materials and results.  "17.500 Euro"  should be "17,500 Euro". On page 5. Is S* the adapted MCS score? On page 7.  The conclusion needs to be revised with brief remarks.

Author Response

Dear Reviewer,

thanks for you very much for your helpful feedback. Please find our response to your comments below.

The abstract needs to be improved, i.e., providing more insights of this paper and more details of the materials and results

We have revised the abstract and in particular added additional information about our data sources and summarizing thoughts on the merits of taking a distributional perspective.

"17.500 Euro"  should be "17,500 Euro". On page 5

We adapted the figure as recommended.

Is S* the adapted MCS score? On page 7.

We have revised the manuscript and below equation (2) in line 208 it is now stated that S* denotes the transformed MCS scores to avoid any potential confusion.